# Geometric morphometric analysis of morphologic disparity, intraspecific variation and ontogenetic allometry of beyrichitine ammonoids

Eva Alexandra Bischof[1]*, Nils Schlüter[2], Jens Lehmann[1]

**1** Geowissenschaftliche Sammlung, Fachbereich Geowissenschaften, Universität Bremen, Bremen, Germany, **2** Naturhistorisches Forschungsinstitut, Museum für Naturkunde, Humboldt-Universität, zu Berlin, Berlin, Germany

\* eva.bischof@hotmail.com

## Abstract

Beyrichitine ammonoids of NV Nevada reveal a high taxonomic diversity of Anisian (Middle Triassic). This diversity is, however, in contrast to their relatively low morphologic disparity. Depending on the exact definition, morphologic disparity of a data set is a direct consequence of the sum of all ontogenetic changes. In the past, however, the interplay of both morphological processes has only rarely been addressed. Using geometric morphometric methods, this study aims at a quantification of allometric processes and the morphologic disparity of beyrichitine ammonoids. The multivariate statistical analysis revealed that morphologic disparity, intraspecific variation respectively, within and between the studied species seems to be the result of deviations in the ontogenetic allometric growth pattern (i.e. heterochrony). During deposition of the studied stratigraphic sequence, a general progressive pedomorphism (juvenilization) was observed. The intraspecific variability pattern coincides with the total morphologic disparity of the analyzed species, which suggests that intraspecific variability facilitated morphologic disparity. The comparison of ontogenetic allometric patterns and changes in intraspecific variation and morphologic disparity are likely to refine our understanding of the intrinsic factors influencing the speciation of this group.

## Introduction

Analysis of morphology and ontogeny are the source for evolutionary and developmental studies in deep time. Since the late 20th century, developmental concepts such as heterochrony [e.g., 1, 2–7] and morphological disparity and intraspecific variation [e.g., 8, 9–12] have proven to be an invaluable source of information complementing taxonomic approaches [10, 13–15] and enriching our knowledge of evolutionary dynamics [8, 13, 14, 16, 17]. Morphological disparity and intraspecific variation can be regarded on a hierarchy level. Morphological disparity is the quantification of morphological variation among species and higher taxa considers the variation within species [18]. If disparity is regarded as differences between ontogenetic end

**Data Availability Statement:** The data is supplied in the Supporting Information (S2).

**Funding:** This research received support from the German Science Foundation (DFG), project

"Nevammonoidea"' (LE 1241/3-1). The funders had no role in study design, data collection and analysis, decision to publish, or preparation of the manuscript.

**Competing interests:** There are no competing interests.

points, it can be argued that the total amount of disparity is a direct consequence of hetero- chrony *sensu* Alberch 1979 [16]. Knowledge of the ontogenetic trajectories is therefore a pre- requisite for enlarging our understanding of macroevolutionary development [13]. However, despite their close relationship, morphologic disparity patterns have been relatively rarely addressed in the context of heterochronic analyses [e.g., 10, 16, 19–23].

In general, heterochrony is defined as change in timing (age) or rate (size) of development relative to the ancestor [24]. However, changes in rate and timing of ontogenetic events can, by definition, only be determined where the age of compared individuals is known [2; chap. 2]. Par- ticularly in paleontology, exact growth rates are often not known [25, 26]. In ammonoids the accretionary growth with preservation of previous chambers, adds a relative time component to the analysis (i.e. the more whorls an individual has, the older it must be). Here it was assumed that the analyzed species have similar coiling rates (i.e., the individual species develop the same number of whorls in the course of their life). Where growth rate is similar between two groups, allometric relationships may reflect true heterochronies [2; chap. 2]. In general, ammonoids are ideal model organisms to study ontogenetic change, intraspecific variability, and macroevolu- tionary patterns [27–30]. That is reasoned in their wide paleogeographic distribution, high pres- ervation potential and high taxonomic diversity and morphological disparity [31].

Previous research has shown that late Anisian (Middle Triassic) ammonoid assemblages of the Fossil Hill Member in NV Nevada are well suited as morphological case studies [14]. The ammonoids are very abundant and well preserved almost throughout the member. In addi- tion, the continuous thin-layered calcareous successions were deposited in a rather stable and calm paleoenvironment [32]. During the depositional period of the Fossil Hill Member no major paleoenvironmental shifts were detected. Therefore, the sequences allow to trace mor- phological change of the species on a high-resolution stratigraphical scale.

Previous [33] geometric morphometric analyses on the Anisian family Ceratitidae Mojsisovics, 1879 of Nevada revealed that members of this group cover a wide range of taxonomic diversity, which, however, is associated with rather low levels of morphologic disparity [14]. Quantification of morphological disparity, intraspecific variation and ontogenetic allometry would provide fur- ther insight into the evolutionary history of these species. However, by definition, phylogenetic information is crucial in determining heterochronic changes among taxa [2; chap. 2]. To our knowledge, for ceratitid ammonoids there is no phylogenetic framework available. For the analy- ses herein, we therefore focused on more closely related species of the subfamily Beyrichitinae Spath, 1934 [34]. The continuous successive stratigraphic sections of the Fossil Hill Member of NV Nevada allow to trace the development of individual species of this subfamily.

In this study, we used a landmark-based geometric morphometric approach including a suite of multivariate statistical tests to study morphological change through ontogeny of beyri- chitine ammonoids. In a first step, we analyzed ontogenetic allometric trajectories and com- pared the individual patterns to each other at the species level. The trajectories were then used to investigate whether intra- and interspecific variability patterns of whorl shape can be noticed among all studied species. The analysis of such patterns might reveal important evolu- tionary mechanisms in the diversification of this clade. In a second step, we contrasted the concept of heterochrony, intraspecific variation and morphologic disparity in order to assess their interlinkage.

## Material & methods

### Fossil material

For this study 46 specimens representing a total of eight species of the genera *Gymnotoceras* Hyatt 1877 [35], *Frechites* Smith 1932 [36] and *Parafrechites* Silberling & Nichols 1982 [37]

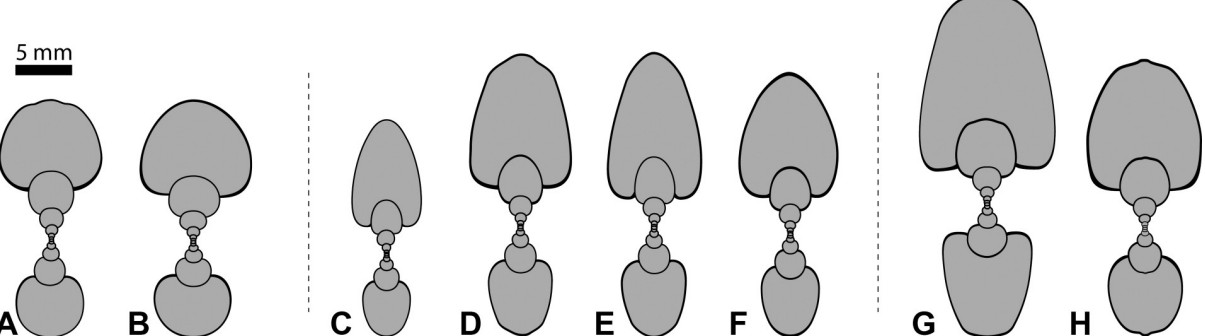

**Fig 1. Cross sections of the analyzed ammonoid species of the late Anisian Fossil Hill Member of NW Nevada, USA; all × 1.5.** (A) *Frechites nevadanus*, GSUB C12377. (B) *Frechites nevadanus*, GSUB C13250. (C) *Gymnotoceras weitschati*, GSUB C14190. (D) *Gymnotoceras blakei*, GSUB C12243. (E) *Gymnotoceras mimetus*, GSUB C13814. (F) *Gymnotoceras rotelliformis*, GSUB 11594. (G) *Parafrechites dunni*, GSUB C9356. (H) *Parafrechites meeki*, GSUB C12536.

were analyzed. These three genera all belong to the subfamily Beyrichitinae Spath, 1934. The selected species are the most characteristic ammonoids of the late Anisian Fossil Hill Member of NW Nevada, USA. All specimens were collected at the Fossil Hill of the Humboldt Range and in the Muller Canyon of the Augusta Mountains (Pershing County), north-western Nevada, USA. The material is stored in the Geosciences Collection of the University of Bremen (GSUB), Germany. More information on the meta data of the fossil material (i.e. geological framework, biostratigraphy, morphologic comparison, data acquisition) can be found in [14]. The U.S. Department of the Interior, Bureau of Land Management (BLM, Nevada State office, Winnemucca District) gave permission to collect samples in the Wilderness Study Area of the Augusta Mountains and P Embree (Orangevale, CA, USA) gave allowed to conduct field work at the fossil Hill locality (private property).

Members of the three genera in focus are morphologically similar [14, Figs 3–5]. The studied species belong to the discoidal morphospace and mainly differ through characteristic ways of ventral arching and some gradual differences in ribbing (Fig 1). Bischof, Schlüter (14) have shown that all species follow the same ontogenetic pathway from very depressed to compressed whorls. However, while some species complete the entire path, some stop their development at an earlier ontogenetic stage. This study therefore examines the closely related species in more detail and quantifies the allometric differences between them.

**Data acquisition.** The underlying shape data of this study represents an excerpt of the data of Bischof, Schlüter (14). The landmarks were retrieved using tpsDig2 v.2.31 [38]. Every half whorl is a separate configuration, which is represented by 16 landmarks (Fig 2). The set of landmarks consists of two unpaired (1, 2) and seven pairs of landmarks (3–16), of which eight are sliding semi-landmarks. To omit missing values in subsequent analyses, the data set was limited to 11 half volutions (i.e., half whorl or growth stage number 5.5). Therefore, every specimen is represented by a total of 176 landmarks (16 landmarks on 11 half whorls).

**Geometric morphometric analysis.** The exact appearance of any morphospace depends on the shapes which are being analyzed. Therefore, only the raw data of Bischof, Schlüter (14) were used and the basic geometric morphometric analysis was repeated as briefly summarized below in this sub-section. The adapted R script and the landmark data are provided in S1 and S2 Files. All geometric morphometric analyses were done using the R software v 3.6.3. [R Core 39] including the R packages *Morpho* v2.8 [40], *geomorph* v3.3.1. [41] and *RRPP* v0.6.0 [42, 43].

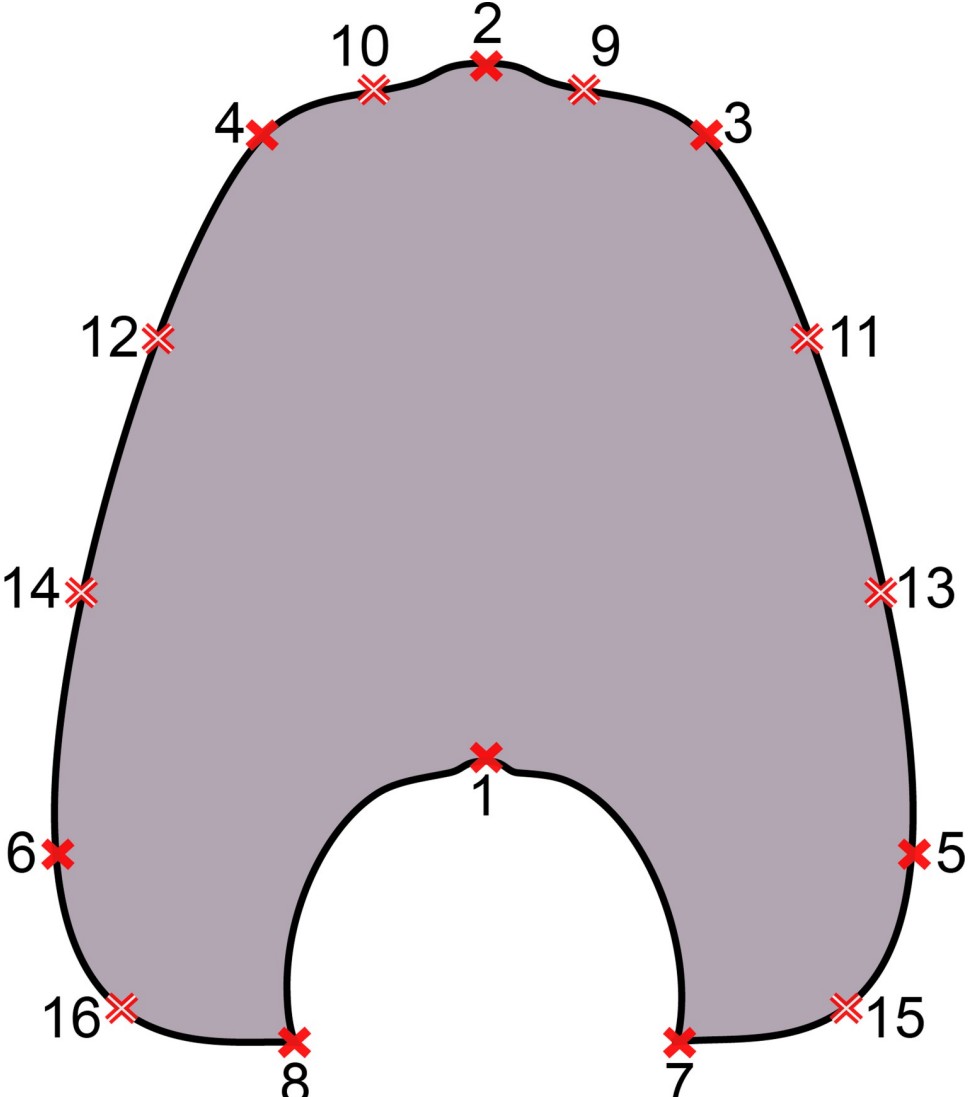

**Fig 2. Digitized sketch of high-precision cross-section of an ammonoid whorl with position of landmarks.** Filled crosses: fixed landmarks; empty crosses: sliding landmarks; black numbers: number of landmark. Definition of fixed landmarks: 1) venter of preceding whorl; 2) venter of whorl; 3 and 4) ventral shoulder or point of highest curvature; 5 and 6) maximum width; 7 and 8) Umbilical seam.

In a first step the 2D landmark coordinates were subjected to a full generalized Procrustes alignment (GPA) using the *Morpho*::*procSym* function. The full Procrustes fit standardizes size, orientation and position, leaving only the Procrustes shape coordinates [44]. Subsequently, the R function *geomorph*::*combine.subsets* was used to standardize all configurations of all growth stages to unit centroid size ($\log_{10}CS$; un-weighted Procrustes shapes). Centroid size is regarded as a proxy for the size of the whorls and equals the square root of the summed squared distances of each landmark from the centroid of the landmark configuration before the GPA [3]. For the calculation of the two-dimensional morphospace we then ran a principal component analysis (PCA) on the aligned un-weighted Procrustes shape coordinates.

To illustrate morphologic variation an ontogenetic trajectory space and a developmental morphospace were calculated. In an ontogenetic trajectory space, the ontogenetic trajectory of

every individual is reduced to a single data point [14, formerly called ontogenetic morphospaces, 45]. The difference of allometric spaces *sensu* Gerber, Eble (13) to trajectory spaces is that the first use allometric trajectories of the size-shape space instead of ontogenetic trajectories. Whereas ontogenetic trajectory spaces are a tool to examine if the ontogenetic pathways of individuals differ, developmental morphospaces *sensu* [21, p. 40] describe how trajectories vary (chapter 4.2 and 4.3 of this thesis, formerly called ontogenetic morphospaces). Developmental morphospaces *sensu* Eble (21, p. 40) directly reflect developmental processes. In the context of this study this means that every individual dot in the developmental morphospace is represented by a specific ontogenetic stage (i.e. half whorl) of an individual [14]. The extreme shapes of the developmental morphospace, were calculated with the R function *GeometricMorphometricsMix::reversePCA* [46] and for the computation of the thin-plate spline deformation grids the R function geomorph::plotRefToTarget was used. All scatterplots were drawn with the R package ggplot2 [47].

**Ontogenetic allometry.** Depending on the scientific context, there are various, closely related biological and evolutionary developmental (evo-devo) terms that can be associated to studies of the size-shape space: Allometry [e.g. 13, 48, 49], allometric space [e.g. 13, 17] or simply heterochrony [e.g. 2, 4, 24]. To unravel patterns in the ontogenetic development of individuals and the phylogenetic variation between taxa, the size-shape relationship was analyzed here. In this study the allometric space was calculated with a regression analysis of the log-transformed size on the values of the first primary component (PC1) of each species (PCshape ~ $\log_{10}$CS*Group) using the R function *geomorph::procD.lm* with 999 iterations and Sum of Squares type I. To quantify the relationship between Procrustes shape variables and a predictor (here log-transformed centroid size) the function fits a linear model to the Procrustes data and creates a size-shape space. A significant association between size and shape for particular species rejects the null hypothesis of isometry (no change in shape during growth) and reveals the influence of allometry.

**Quantification of allometric trajectories.** To quantify interspecific and intergeneric allometric relationships, we performed a homogeneity of slopes (HOS; R function *RRPP::pairwise*) test and a phenotypic trajectory analysis (R function *RRPP::trajectory.analysis* with 999 iterations). Both functions test for differences in the slope angle and length of the shape-size relationship to quantify the amount of shape variation, which is explained by size. More details on these procedures can be found in Esquerré, Sherratt ([50], p. 2832).

Allometry is associated with changes during growth with age by definition. Since the age of fossil specimens can only hardly be determined, size and whorl number are used here as a proxy for age. Whereas the HOS test regards size as a continuous variable (i.e. $\log_{10}$size), the *trajectory.analysis* function depends on at least one categorical interaction variable (i.e. number of whorl) as a proxy for size. Therefore, the above computed allometric model (PCshape ~ $\log_{10}$CS*Group) was used in the HOS test and in the trajectory analysis a linear model including the whorl number (PCshape ~ Group * WhorlStage) was used. In general, differences in slope angles and maximum centroid size can be the result of heterochronic processes. For the terminology for heterochrony we follow the concept of [4] which is nicely illustrated in [24].

**Morphologic disparity and intraspecific variation.** To examine how intraspecific variability and morphological disparity changes through the development of individuals, we used the R function *geomorph::morphol.disparity*. Intraspecific variance is calculated in an analogous manner to morphological disparity [3]. The function calculates absolute differences in Procrustes variances of specified groups and tests for pairwise differences in Procrustes variances between these groups while accounting for group size. The statistical significance of the calculated Procrustes variances between the different growth stages was assessed using a randomized residual permutation test with 999 iterations.

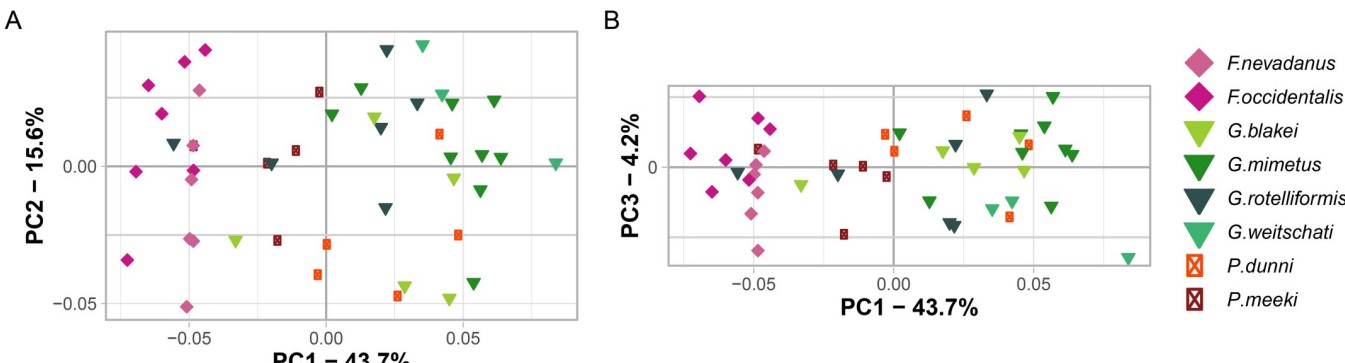

**Fig 3.** PCA plots of ontogenetic trajectory space of all species of (A) Principal Component 1 and 2; (B) Principal component 1 and 3.

It is very important to note that only discontinuous predictors (i.e. whorls) can be used here. In case a continuous predictor such as $\log_{10}$CS is used, the function *morphol.disparity* uses the overall mean of all configurations.

## Results

### Ontogenetic and developmental morphospaces

The ontogenetic (Fig 3) and developmental (Fig 4) morphospace of beyrichitine ammonoids do not substantially differ from the ceratitid morphospaces that were described in Bischof, Schlüter [14]. The first three components (PCs) of the Principal Component Analysis (PCA) on the data set with joint configurations (i.e. ontogenetic trajectory spaces, Fig 3) account for 63.5% ($PC_1 = 43.7\%$, $PC_2 = 15.6\%$, $PC_3 = 4.2\%$) of the total variation. Since most of the variation is explained by the first component, the placement of the species along PC1 is the most important characteristic. Whereas the genus *Frechites* occupies the left side (low PC 1-values), in the central and right part of the diagram *Gymnotoceras* and *Parafrechites* intermingle to some extent. This distribution is a first indication that the ontogenetic trajectories have taxonomic significance.

The developmental morphospace (Fig 4) shows the same three basic ontogenetic phases as described in Bischof, Schlüter [14]: (1) Earliest whorls are flattened and occupy the lower left quadrant (negative PC1 and PC2 values; Fig 5A); (2) the depressed whorls of juveniles cover the central area of plot (PC1 = 0; PC2 slightly positive; Fig 5B); (3) adults have more compressed and stout whorls, right side of the diagram (positive PC1 and negative PC2 values; Fig 5C). The first three components of the PCA of all growth stages of the specimens as separate configurations (i.e. developmental morphospace) account for 93.8% ($PC_1 = 80.1\%$, $PC_2 = 10.7\%$, $PC_3 = 3.0\%$) of the total variation.

Since the herein studied species show Type A and Type B ontogenetic trajectories (for a discussion on types see Bischof et al. 2021), there are two extreme adult shapes (Fig 6): Type A) rather depressed, stout conches that do not overlap much the preceding whorl and are associated with much shorter ontogenetic trajectories (*F. nevadanus*, *F. occidentalis*, *P. meeki)*, and Type B) compressed conches with a more pronounced venter and a higher degree of overlap with the preceding whorl (*G. blakei*, *G. mimetus*, *G. rotelliformis*, *G. weitschati*, *P. dunni*).

### Examining ontogenetic allometry: Regression model

Whorl shape is influenced by the centroid size, the group (species and/or genus) and the interaction of the two (R function *geomorph*::*procD.lm*; Table 1). This indicates that there is an

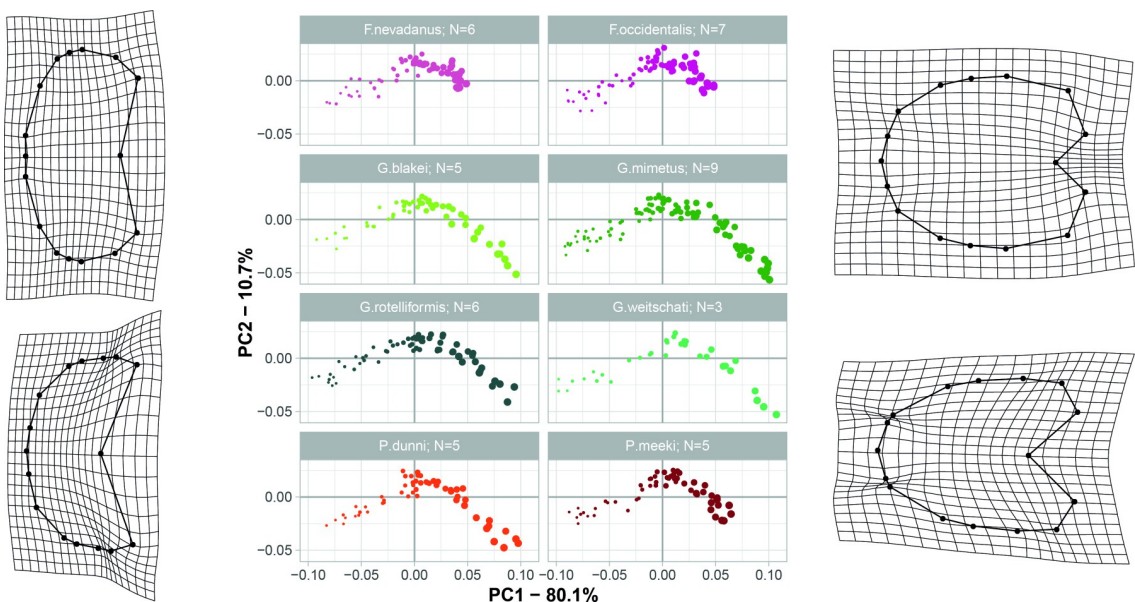

**Fig 4. Developmental morphospace with PCA of un-weighted Procrustes shape variables.** Point size refers to growth stage. Deformation grids show the transformation of the mean shape to the modeled shapes of the extreme values for PC1 and PC2.

ontogenetic allometric (and not isometric) pattern present and the allometric slopes differ amongst species and genus. The full model (PCshape ~ $\log_{10}$CS * species; Table 1) describes the shape significantly better than the reduced model (PCshape ~ $\log_{10}$CS; S3 File). Therefore, the full model was used for further analysis.

The fitted values of the linear regression model were plotted against the $\log_{10}$CS (Fig 7). Differences in slopes of the ontogenetic allometric trajectories are caused by changes in ontogenetic allometric patterns. There is not much variation in shape of whorl 0.5 between the individual species (Figs 5 and 7), but the range of different shapes of whorl 5.5 is more complex (Figs 6 and 7). The species *G. weitschati* displays a much smaller y-intercept (Fig 7) than all other species. However, with three specimens this species had the smallest sample size (Fig 4) and therefore also less statistical power.

TPS splines of Procrustes shapes for extreme regression values against the mean shapes of respective growth stage. The values of the y-intercept and the slopes of all species can be found in Table 2.

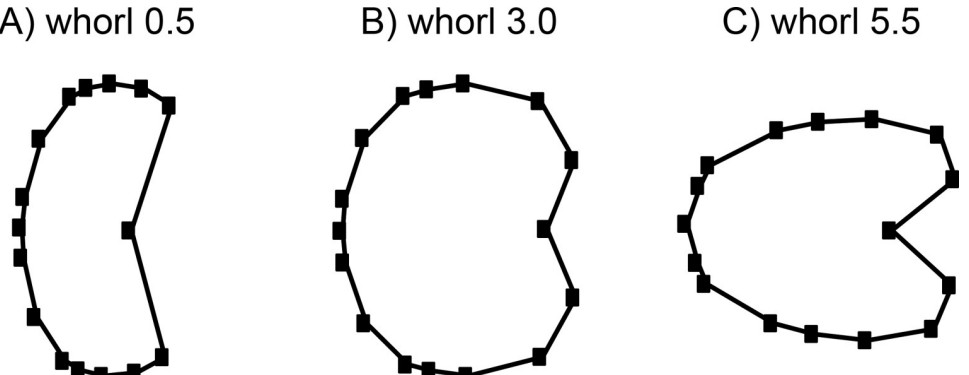

**Fig 5. Mean shapes of whorl stages.** (A) Whorl stage 0.5, (B) whorl stage 3.0 and (C) whorl stage 5.5.

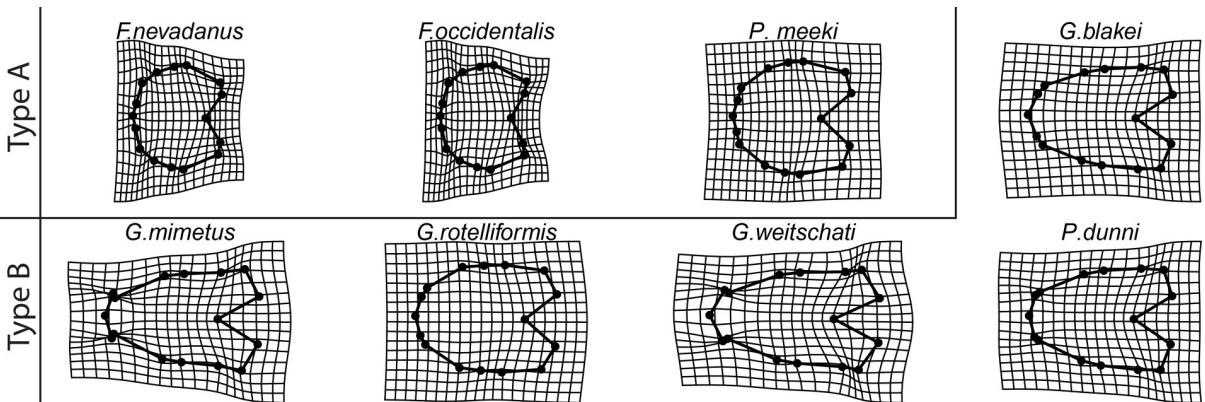

**Fig 6.** TPS spline of mean shape of whorl 5.5 of all species in this study (grey) plotted against the mean shape of whorl 5.5 of the respective species (black).

## Evaluation and quantification of allometric slopes

Allometric slopes differ among species (Fig 7) and genera as well. For the statistical quantification of the allometric trajectories an allometric trajectory analysis (R function *RRPP:: trajectory.analysis*) and a homogeneity of slopes test (HOS; R function *RRPP::pairwise*) were performed. Pairwise comparison of allometric trajectories revealed that all genera and most species have significantly different pairwise slopes (direction of shape change) in at least one of the two test procedures (see S4 File). Three species pairs did not reveal statistically significant slopes in either of the two tests (*G. blakei-P. dunni*, *G. weitschati-G. mimetus* and *G. weitschati-G. rotelliformis*). The members of the non-significant pairs all belong to type B.

Whereas the minimal centroid size does not vary much between the individual species, there is more variability in later ontogenetic stages. In general type A species attain smaller maximum (centroid) sizes than type B species. Accordingly, the prevailing heterochronic process from type B to type A can be regarded as a neoteny (pedomorphosis).

**Table 1. Analysis of variance table obtained from the Procrustes ANOVA (R function RRPP::lm.rrpp) of the full model (PCshape~ log10CS * group) on the species and genus level.**

| Species | Df | SS | MS | $R^2$ | F | Z | Pr(>F) |
|---|---|---|---|---|---|---|---|
| log.size | 1 | 0.98264 | 0.98264 | 0.70889 | 1463.5398 | 7.2528 | 0.001 |
| species | 7 | 0.02765 | 0.00395 | 0.01995 | 5.8834 | 6.1831 | 0.001 |
| log.size:species | 7 | 0.04689 | 0.00670 | 0.03383 | 9.9769 | 7.7669 | 0.001 |
| Residuals | 490 | 0.32899 | 0.00067 | 0.23734 | | | |
| Total | 505 | 1.38617 | | | | | |
| **Genus** | **Df** | **SS** | **MS** | $R^2$ | **F** | **Z** | **Pr(>F)** |
| log.size | 1 | 0.98264 | 0.98264 | 0.70889 | 1401.3677 | 7.2141 | 0.001 |
| genus | 2 | 0.01675 | 0.00837 | 0.01208 | 11.9416 | 4.9199 | 0.001 |
| log.size:genus | 2 | 0.03619 | 0.01809 | 0.02611 | 25.8034 | 6.4041 | 0.001 |
| Residuals | 500 | 0.3506 | 0.0007 | 0.25293 | | | |
| Total | 505 | 1.38617 | | | | | |

Df: Degrees of freedom; SS: Sum of squares; MS: mean squares; $R^2$: coefficient of determination; Z: Effect sizes (Z) based on F distributions; Pr(>F): p-value of F statistics.

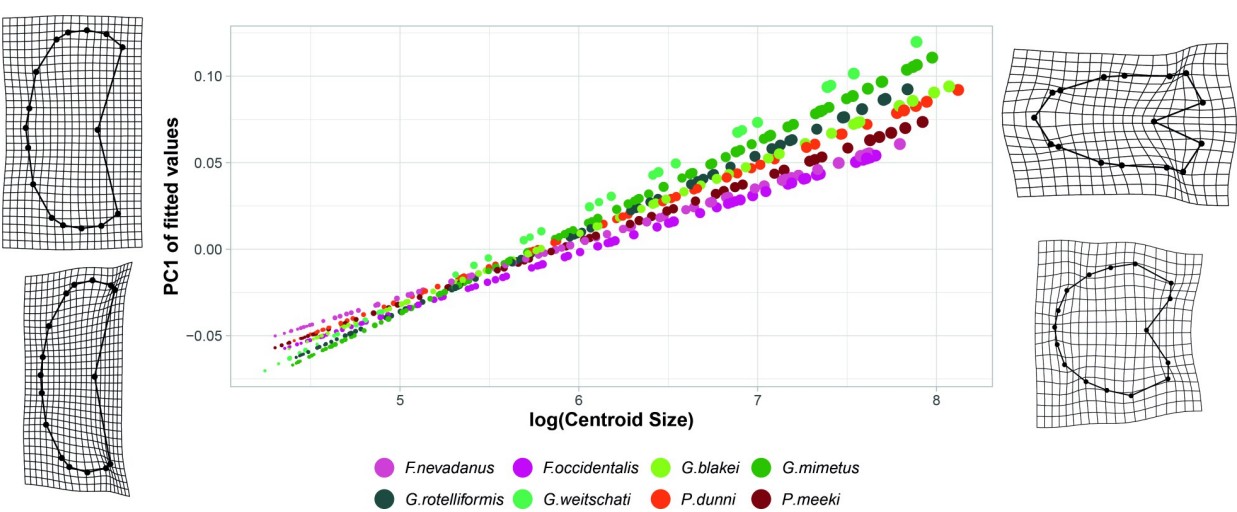

**Fig 7. Ontogenetic allometric trajectories with fitted values of the linear regression model plotted against log$_{10}$CS.** Point size refers to growth stage.

## Intraspecific variation and morphological disparity during ontogeny

A general pattern in the levels of intraspecific Procrustes variance (Fig 8A) among the whorls can be observed in all investigated species: a faint decrease of relative variability occurs after the first whorl (0.5), with a general slight increase after whorl stage 1.5 or 2.0. The taxa *F. nevadanus*, *F. occidentalis* and *P. meeki*, have a relatively homogenous pattern with no significant differences among the whorls in the levels of Procrustes variance. In *G. weitschati*, *G. mimetus*, *G. blakei*, *G. rotelliformis* and *P. dunni*, in contrast, the Procrustes variance of whorl 5.5 is approximately doubled to tripled compared to previous whorls (see. Fig 8A for visual comparison, S5 File for data). Whereas the Procrustes variance of whorls 0 to 5.0 of *G. blakei*, *G. rotelliformis* and *P. dunni*, are again relatively homogenous, as in above mentioned species, *G. mimetus and G. weitschati* are the only species that have a second significant minima of variance between the whorl stage 3.0 and 5.5.

The total variance of the respective whorls (Fig 8B) of the investigated species is relatively homogenous among whorls 0.5–5, with a relatively small drop from whorl 0.5 to whorl 1.5. The variance of following whorls is generally increased again, though, admittedly, no significant differences among whorls 0.5 –whorl 5 can be reported; only whorl 5.5 with the most elevated level of Procrustes variance differs significantly to previous whorls.

**Table 2. Intercept and slopes of linear model of fitted shape values on centroid size.**

| Species | Ontog. Type | Stratigraphic order | y-Intercept | Regression slope |
|---|---|---|---|---|
| *G. weitschati* | B | 1 | -0.2914 | 0.05211 |
| *G. mimetus* | B | 2 | -0.2852 | 0.04963 |
| *G. rotelliformis* | B | 3 | -0.2626 | 0.04529 |
| *G. blakei* | B | 4 | -0.2374 | 0.04107 |
| *P. dunni* | B | 7 | -0.2247 | 0.039 |
| *P. meeki* | A | 6 | -0.2119 | 0.03603 |
| *F. occidentalis* | A | 8 | -0.2042 | 0.03372 |
| *F. nevadanus* | A | 5 | -0.1867 | 0.03175 |

Stratigraphic order from 1 = old to 8 = young.

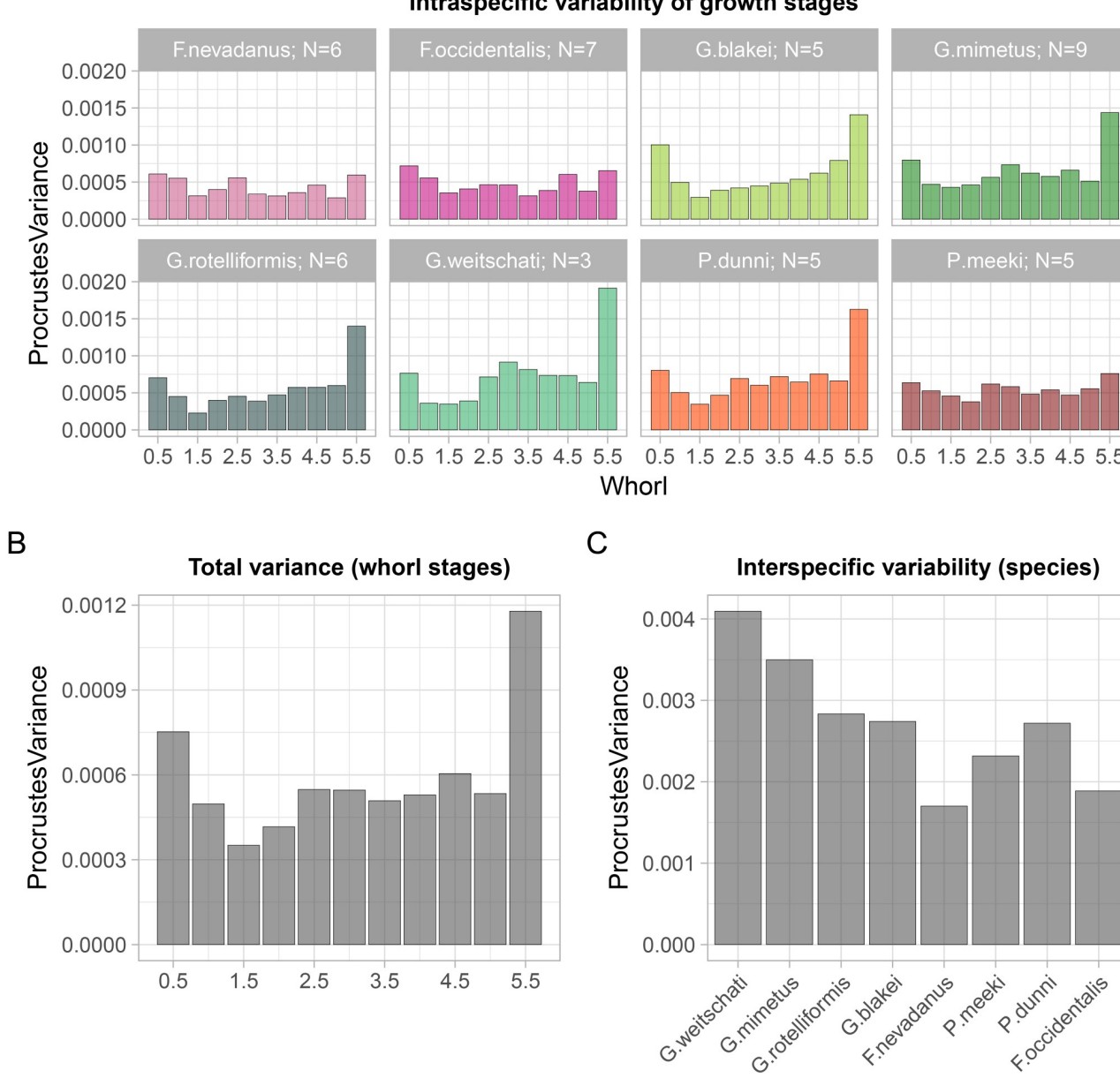

**Fig 8. Amount of procrustes variance of intraspecific variation and morphologic disparity of whorl stages and species.** (A) Intraspecific variability patterns broken down by whorl stage. Contribution of individual whorls of species to intraspecific variation. The x axis shows the growth stages from 0.5 to 5.5. (B) Total Procrustes variance (i.e. morphological disparity) of all species at specific growth stages. The x axis shows the growth stages from 0.5 to 5.5. (C) Contribution of species to overall intraspecific variation. Species are ordered in stratigraphic order from left-old to right-young.

The total intraspecific variation (Fig 8C) in stratigraphic order shows a downward trend from old to young. Especially within the evolutionary line of the genus *Gymnotoceras* the intraspecific variation is constantly decreasing.

## Discussion

In this study, ontogenetic allometric trajectories and intra- and interspecific variability patterns of ammonoids were quantified for the first time using geometric morphometric

methods. The methods used represents an extension of the methods previously introduced by Bischof, Schlüter [14]. The analysis of the evolutionary patterns revealed that the analyzed species do not really differ through morphological shapes that are developed *per se*, but rather through their individual developmental rates, i.e. heterochrony (Figs 4 and 8). This is confirmed by differences in ontogenetic (Fig 3) and allometric ontogenetic trajectories (Fig 7) among species. The ontogenetic development of the species in focus are characterized by several changes in intraspecific variability (Fig 8) and developmental rate or "timing" of whorl outline (Fig 7; Table 1). These findings go in line with previous research on ceratitid ammonoids by Bischof, Schlüter (14).

## Patterns in ontogenetic allometry

The ontogenetic development of the species and genera in focus are characterized by changes in developmental rate or "timing" of whorl shape (Fig 7; Table 1), which manifests itself in an almost constant decrease of the allometric slope in stratigraphic order (Table 2). In general, adult representatives of stratigraphically younger species resemble juveniles of stratigraphically older individuals, i.e. pedomorphosis or juvenilization [2, p. 14, 4, 5, p. 209].

Due to their small size (+/- 0.4 mm) the earliest whorls are more difficult to measure than later and larger ontogenetic stages. However, more importantly, this large difference implies also that the relative measurement error associated with the earliest (smaller) ontogenetic stages is much larger [14]. Such measurement inaccuracies have a greater influence on the y-intercept (more strongly influenced by shape at origin) than on the slope of the allometric trajectories (reflects interplay of all shapes during ontogeny). Therefore, it is suggested that pre- and/or post-replacement of the allometric trajectories was induced by measurement inaccuracies. The full ontogenetic sequence of the studied specimens, as already described by Bischof, Schlüter [14], includes the transition from flat to depressed to compressed whorls. Thereby the studied species can be divided in two main ontogenetic groups: B) longer trajectories that refer to an elongated adult whorl shape (*G. blakei*, *G. mimetus*, *G. rotelliformis*, *G. weitschati*, *P. dunni*), and A) truncated trajectories that are caused by "juvenilized" (pedomorphic), rather rounded and stout adult whorls (*F. nevadanus*, *F. occidentalis*, *P. meeki*). Regarding the quantification of allometric heterochrony the following three main statements can be made: (1) The pairwise difference in slope of the individual species and genera are statistically significant. (2) The evolutionary lineage of *Gymnotoceras* lineage is characterized by a progressive decrease in allometric slope, i.e. neoteny (pedomorphosis). (3) The smaller allometric slope of Type A species compared to Type B species is indicative for neoteny (resulting in pedomorphosis).

It is not uncommon for ammonoids to show attenuated or hyper-adult morphological characteristics even within a single species (from a single stratigraphic interval) [51]. One of the first studies describing intraspecific variation through ontogeny was conducted by Schmidt in 1926 [7]. He analyzed intraspecific variability patterns in Carboniferous ammonoids within single stratigraphic layers and introduced the terms bradymorphic (retention of juvenile characteristics) and tachymorphic (hyper-adult characteristics) to refer to morphologic end members of a species. It must, however, be noted, that intraspecific variation of the investigated species in this study does not arise from brady- nor tachymorphic processes.

In the material studied, as well as in most other ammonoid assemblages, pedomorphosis is the dominant heterochronic process [52–54]. According to [52] the general prevalence of pedomorphosis in ammonoid evolution may reflect both a real phenomenon as well as the fact that this type of heterochrony is easier to detect. A more courageous interpretation is that certain heterochronic changes would be adaptively favored in particular environments [55]. Thereby, more stable environments can rather be associated with slowed down growth of the

pedomorphic development (Neoteny) [2, chap. 4, 5, chap. 8]. The relatively calm and stable paleoenvironment under which the successions of the Fossil Hill Member were deposited [32] do not challenge this picture. In this context, it would be of greatest interest to apply the methods used here to ammonoid assemblages that have been exposed to more palaeoenvironmental pressure.

## Intraspecific variability and morphological disparity

Two unequivocal patterns of intraspecific variability during ontogeny can be recognized among the studied species (Fig 8). The first pattern is characterized by homogeneous levels of Procrustes variance among the whorls and can be detected in *F. nevadanus*, *F. occidentalis* and *P. meeki* (ontogenetic type A). The ontogenetic trajectories of these species are shorter (Fig 4), i.e. reflect pedomorphic development. The second pattern shows a significantly increased level of Procrustes variance in the last whorl (whorl 5.5). The latter pattern is particularly seen in ontogenetic type B species: *G. weitschati*, G. *blakei*, G. *rotelliformis*, G. *mimetus* and *P. dunni*. Similar developmental patterns were detected by Gerber, Neige (56; tab 2; Fig 6) that described significant higher levels of variance of adult hildoceratid ammonoids than their corresponding juveniles. By applying traditional morphometric approaches, [12, 56] studied intraspecific variation during the ontogeny in ammonoids, resulting in no clear patterns in variation. The differences in study design hinder a close comparison of their study with ours.

In general, there is no clear-cut pattern in the variability during ontogeny in ammonoids. Studies report ontogenies for instance with a decrease in variability from oldest to youngest developmental stages and vice versa [16, 51]. The intraspecific variability patterns of the studied species can be explained by their ontogenetic developmental grouping rather than by their taxonomic assignment. Particularly within the evolutionary line of the genus *Gymnotoceras* the total Procrustes variance (Fig 8C) is characterized by a progressive decrease in stratigraphic order (from old to young).

In case of the studied material elevated levels of intraspecific variation coincide with the occurrence of the three "extreme" shapes (flat–depressed–compressed; Fig 5). When transitions between developmental stages are accompanied by abrupt changes [critical points or "Knickpunkte"; 57, 58], developmental disparity patterns are likely to be polyphasic as well [59]. It is remarkable that in the intraspecific variability of particularly the ontogenetic type B group (Fig 8A; *G. weitschati*, *G. blakei*, *G. rotelliformis*, *G. mimetus* and *P. dunni*) a pattern is revealed which coincides with the total interspecific morphologic disparity of the analyzed species (Fig 8B). The variance of the latest whorl (5.5) is significantly elevated, in comparison to the remaining whorls, in either both, morphologic disparity and intraspecific variability, patterns. Accordingly, the shape of whorl 5.5, which yields the highest disparity in the analyzed species, has also the highest level of intraspecific variability. This might indicate weakened developmental constraints that promote disparity in the shape of the latest whorl among the analyzed beyrichitine ammonoids. Similar observations were made in species of the Late Cretaceous echinoid *Micraster* [60].

## Conclusion

Even though heterochronic processes are possible factors responsible for intraspecific variation and morphologic disparity, the interplay of heterochrony and morphological disparity has only rarely been addressed [16]. In this study heterochronic relationships and morphologic disparity of ammonoids were quantified using geometric morphometric methods on ontogenetic cross-sections for the first time. Comparisons of disparity, when combined with

multivariate statistical analyses of ontogenetic allometric trajectories, help to quantitatively assess the role of development in shaping morphospace occupation and adult disparity [13].

The geometric morphometric analysis of this study revealed that intraspecific variability patterns of the studied species are only roughly linked to their taxonomic classification; rather it is explained by their individual ontogenetic developmental grouping. Therefore, intraspecific variation and morphologic disparity in beyrichitines seems to be the result of perturbations of the allometric growth pattern (i.e. heterochrony). There is evidence that more stable environments can generally rather be associated with slowed down growth of pedomorphic development, i.e. neoteny [2, chap. 4, 5, chap. 8]. This suggests that the rather calm and stable environment of the Fossil Hill Member [32] favored pedomorphic processes, which is reflected by the continuously decreasing allometric slopes (Fig 7) of the species studied here.

The comparison of ontogenetic allometry patterns and changes in morphologic disparity are likely to refine our understanding of the intrinsic factors influencing the speciation of this group. Even if the methods introduced herein might not supply the full causal explanation on the diversity and disparity patterns observed, they offer additional insights into the macroevolutionary processes in ammonoids. This study therefore underlines the importance of using quantitative multivariate analyses to properly assess the role of ontogenetic processes in shaping morphologic disparity across species.

## Supporting information

**S1 File. R script for geometric morphometric analysis.**
(R)

**S2 File. Landmark data of analyzed species (retrieved with tpsDig2).**
(TPS)

**S3 File.**
(XLSX)

**S4 File.**
(XLSX)

**S5 File.**
(XLSX)

## Acknowledgments

We would like to thank D. Kuhlmann (now Messel, Germany) as former technician of the Geowissenschaftliche Sammlung Bremen for field assistance and mechanical preparation of the material collected by our working group. M. Krogmann (Bremen, Germany) is thanked for field assistance as well as his broad support in the artwork for this article. Our gratitude goes to D. Korn (Berlin) for scientific discussion and moral support during the design of this study. We are indebted to K. Boos for the introduction to the R software and further statistical support. P. Embree (Orangevale, CA, USA) is thanked for broad support, including organization of field-campaigns, scientific information and the permission to collect on his private property. Our gratitude goes to the reviewers Marco Balini and Kenneth De Baets and the editor Cyril Charles for their corrections and suggestions that greatly improved the manuscript.

## Author Contributions

**Conceptualization:** Eva Alexandra Bischof, Nils Schlüter, Jens Lehmann.

**Data curation:** Eva Alexandra Bischof.

**Formal analysis:** Eva Alexandra Bischof.

**Funding acquisition:** Jens Lehmann.

**Investigation:** Eva Alexandra Bischof, Nils Schlüter, Jens Lehmann.

**Methodology:** Eva Alexandra Bischof, Nils Schlüter.

**Project administration:** Jens Lehmann.

**Resources:** Jens Lehmann.

**Software:** Eva Alexandra Bischof.

**Supervision:** Nils Schlüter, Jens Lehmann.

**Validation:** Nils Schlüter.

**Visualization:** Eva Alexandra Bischof.

**Writing – original draft:** Eva Alexandra Bischof.

**Writing – review & editing:** Eva Alexandra Bischof, Nils Schlüter, Jens Lehmann.

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
