## [Decision Letter · Decision Letter 0]

21 Dec 2021

PONE-D-21-30107Geometric Morphometric analysis of morphologic disparity, intraspecific variation and ontogenetic allometry of beyrichitine ammonoidsPLOS ONE

Dear Dr. Bischof,

Thank you for submitting your manuscript to PLOS ONE. After careful consideration, we feel that it has merit but does not fully meet PLOS ONE’s publication criteria as it currently stands. Therefore, we invite you to submit a revised version of the manuscript that addresses the points raised during the review process. From comments provided by Reviewer 2, please consider adding few more references to previous works to improve the discussion.

We look forward to receiving your revised manuscript.

Kind regards,

Cyril Charles

Academic Editor

PLOS ONE

3.Thank you for stating the following in your Competing Interests section: 

“There are no competing interests.”

Reviewers' comments:

Reviewer's Responses to Questions

**Comments to the Author**

1. Is the manuscript technically sound, and do the data support the conclusions?

Reviewer #1: Yes

Reviewer #2: Yes

2. Has the statistical analysis been performed appropriately and rigorously? 

Reviewer #1: Yes

Reviewer #2: Yes

3. Have the authors made all data underlying the findings in their manuscript fully available?

Reviewer #1: Yes

Reviewer #2: Yes

4. Is the manuscript presented in an intelligible fashion and written in standard English?

Reviewer #1: Yes

Reviewer #2: Yes

5. Review Comments to the Author

Reviewer #1: The manuscript is about a complex topic, the morphometric study of ontogeny of selected Anisian ammonoid species from Nevada (USA).

The fossil localities are worlwide known for the abundance of ammonoids and their very good preservation. The systematic study of these ammonoid faunas, based on population analysis and traditional approach, is quite recent (Silberling & Nichols, 1982; Bucher, 1992; Monnet & Bucher, 2005), and is a good starting point for the research of the authors.

This manuscript is focused on a specific group of ammonoids, the Beyrichitinae, that is aboundant in the studied succession and provides several index species for the chronostratigraphy of the Anisian on the North American Paleobioprovince. The 8 species selected for this study are the most representative of this group. Ontogeny of these species has never been cross compared.

The quality of the specimens selected for sectioning is very good. The morphometric mesurements have been made with great accuracy and the data analysis is done with up-to-date mathematic techiques, that have never been applied before to Beyrichitinae.

The manuscript is well organized and written in a very very good English. Figures are all of good quality. The results are very well presented and the discussion is well organized. The identification of pedomorphic relationships within north american Beyrichitinae is an important results. Heterochrony in Triassic ammonoid has been rarely described, therefore this manuscript will be a reference for all the future investigations on Triassic.

I do not see any weak points in the manuscript. It would be interesting to compare the sutures line of the studied species, in order to test if the differences in the suture lines are consistent with pedomorphic relationship. As far as I remember, some suture lines have been published by Silberling & Nichols (1982) from Fossil Hill and perhaps Spath (1934), but I have not checked if sutures are available for all the 8 species under study.

This test could be useful because some Tethyan Beyrichitinae, such as Kocaelia Fantini Sestini 1990, show a peramorphic relationship as regard the suture line, with an increasing complication (accelleration) of the indentitions.

Reviewer #2: You present an interesting and novel approach to study intraspecific variation throughout ontogeny in ammonoid whorl cross section and compare it with morphological disparity. I enjoyed reading and reviewing the paper and look forward to seeing the revised version. I just would like to see a little bit more discussion on previous research in this direction as I feel this would place your work in even broader context, widen its scope and further highlight the importance and innovativeness of your approach. The most important points being:

1) The consideration of heterochrony and ontogenetic trajectories when comparing specimens within and between species: this approach has a long tradition particularly in ammonoid research (e.g., Michalsky 1890 and more formally by Schmidt 1926). Studying intraspecific variation throughout ontogeny and differences (or lack thereof) between species has also been studied by various authors (e.g., particularly in Paleozoic ammonoids: Korn and Vöhringer 2004; Korn and Klug 2007; Ebbighausen and Korn 2007; De Baets et al. 2013; but also, Mesozoic ammonoids: Dommergues and Marchand 1986; Gerber et al. 2007, 2008; Gerber 2011).

2) Heterochrony and the environment: you mainly cite McKinney to back up this association, which is classical work, but additional publications have been published particularly targeting ammonoids (e.g., Landman and Geyssant 1993; Gerber 2011). Also, I feel it would be more appropriate to cite the full chapter reference giving credit to author of individual chapters of McKinney 1991 and would also make it easier to follow on which aspects these chapters focus.

3) Bradymorphic versus tachymorphic: Schmidt (1926) introduced these terms to refer to the extreme end members of trajectories within a species which possess characteristics of earlier whorl later (in terms of heterochrony: paedomorphic) or later whorl earlier in development (in terms of heterochrony: peramorphic), respectively. Also, Beznosov and Mitta (1995) used this terminology. I feel it is worth to at least mention these terms and if these could apply to your observations.

4) Changes in the degree of intraspecific variation through ontogeny: you mention that previous studies reported decrease in variability from the oldest to youngest stage and vice versa. The situation might be even more complex as some authors even report high variation particularly in early and late stage or vice versa in intermediate stages further backing your statement is no support for a clear-cut pattern. It would also speak about “life stage” or “ontogenetic stage” in this context/sentence rather than just “stage” to avoid confusion with geological stages.

These and additional points as well as the full citations of the mentioned references can be found in the annotated pdf (or references listed therein).

Please do not hesitate to contact me in case of questions or if something is unclear.

6. PLOS authors have the option to publish the peer review history of their article (what does this mean?). If published, this will include your full peer review and any attached files.

Reviewer #1: **Yes: **Marco Balini

Reviewer #2: **Yes: **Kenneth De Baets

---

## [Author Response · Author response to Decision Letter 0]

16 Jan 2022

I added all answers and comments in the rebuttal letter.

---

## [Editor Report · Decision Letter 1]

21 Jan 2022

Geometric Morphometric analysis of morphologic disparity, intraspecific variation and ontogenetic allometry of beyrichitine ammonoids

PONE-D-21-30107R1

Dear Dr. Bischof,

We’re pleased to inform you that your manuscript has been judged scientifically suitable for publication and will be formally accepted for publication once it meets all outstanding technical requirements.

Kind regards,

Cyril Charles

Academic Editor

PLOS ONE
---

## [Editor Report · Acceptance letter]

27 Jan 2022

PONE-D-21-30107R1 

Geometric Morphometric analysis of morphologic disparity, intraspecific variation and ontogenetic allometry of beyrichitine ammonoids 

Dear Dr. Bischof:

I'm pleased to inform you that your manuscript has been deemed suitable for publication in PLOS ONE. Congratulations! Your manuscript is now with our production department. 

Kind regards, 

on behalf of

Dr. Cyril Charles 

Academic Editor

PLOS ONE